# The influence of health awareness on university students' healthy lifestyles: The chain mediating role of self-esteem and social support

Zeqing Zhang[1,2]*, Haslinda Abdullah[1], Akmar Hayati Ahmad Ghazali[3], Jeffrey Lawrence D'Silva[1], Ismi Arif Ismail[4], Zerui Huang[1,2]

1 Institute for Social Science Studies, Universiti Putra Malaysia, Serdang, Selangor, Malaysia, 2 School of Culture and Communication, Guangdong Business and Technology University, Zhaoqing, People's Republic of China, 3 Faculty of Modern Languages and Communication, Universiti Putra Malaysia, Serdang, Selangor, Malaysia, 4 Faculty of Educational Studies, Universiti Putra Malaysia, Serdang, Selangor, Malaysia

* zhangzqluna@foxmail.com

**Data Availability Statement:** All relevant data are available from the Inter-university Consortium for Political and Social Research (ICPSR) database. The research data can be accessed using the

## Abstract

As university students face increasing pressures in a highly competitive society, unhealthy lifestyles have become a common phenomenon. Health awareness is considered a critical factor in promoting healthy behaviors, yet its mechanism of action on university students' healthy lifestyles, influenced by self-esteem and social support, remains unclear. This study aims to explore the relationship between health awareness (HA) and healthy lifestyles (HL) and to examine the mediating roles of self-esteem (SE) and social support (SS) in this relationship. Based on social cognitive theory, this study initially constructs a theoretical model of the impact of health awareness on healthy lifestyles among university students, introducing self-esteem and social support as mediating variables to further build a chain mediation model. A total of 1,169 participants (554 males and 615 females) completed an online survey. Participants completed the Health Awareness Scale (HAS), the Diet and Healthy Lifestyle Scale (DEVS), the Rosenberg Self-Esteem Scale (RSES), and the Oslo Social Support Scale (OSSS-3). The results show that high levels of health awareness positively influence self-esteem, which in turn positively influences social support, ultimately having a positive impact on healthy lifestyles. Specifically, self-esteem and social support play significant mediating roles in the relationship between health awareness and healthy lifestyles. The findings reveal the pathway by which health awareness influences healthy lifestyles through self-esteem and social support, enriching the theoretical explanation of health behaviors within the framework of social cognitive theory, particularly in the context of university students. Furthermore, the results provide practical guidance, suggesting that in designing interventions to promote healthy lifestyles, emphasis should be placed on enhancing university students' health awareness, self-esteem, and social support systems. This could include mental health education, establishing supportive communities, and fostering activities that promote self-esteem.

following reference: Zhang, Zeqing. The Influence of Health Awareness on University Students' Healthy Lifestyles in Guangdong. Ann Arbor, MI: Inter-university Consortium for Political and Social Research [distributor], 2024-07-07. https://doi.org/10.3886/E207741V1.

**Funding:** The author(s) received no specific funding for this work.

**Competing interests:** The authors have declared that no competing interests exist.

## Introduction

In a highly competitive society, people face increasing levels of stress, especially university students, who often develop unhealthy lifestyles as a result [1]. These students generally experience poor dietary habits, insufficient sleep, sedentary behavior, and a lack of adequate physical exercise. Such lifestyles adversely affect their physical and mental health [2]. Existing research has demonstrated that balanced diets and active lifestyles play an essential role in preventing chronic non-communicable diseases. Maintaining a balanced diet and a positive lifestyle can effectively reduce the risk of these diseases [3]. Adopting healthy lifestyle habits not only improves quality of life but also helps alleviate stress amidst a busy daily routine [4]. Exploring healthy lifestyles among university students can help enhance their overall health and reduce future health problems.

Therefore, this study aims to examine the influence of health awareness on university students' healthy lifestyles and analyze the chain mediating roles of self-esteem and social support in this process. This study seeks to uncover how health awareness, self-esteem, and social support collectively affect the health behavior patterns of university students, providing theoretical and empirical support for promoting healthy lifestyles among this population.

## Literature review

Health Awareness is defined as the extent of an individual's knowledge and concern about their health status and related behaviors. It should not only focus on health behaviors themselves but also on the individual's attitudes toward health. Measured through self-assessment of health awareness, personal responsibility, and health motivation, health awareness encompasses specific health behaviors and the psychological states that these behaviors may induce [5]. University students generally lack health awareness; despite being in a peak period of physical health, they show relatively low concern for diseases and health. At the same time, factors such as an accelerated pace of life, rapid changes in the external environment, and diverse schedules pose numerous challenges to maintaining a healthy lifestyle [6]. Research shows that an individual's health awareness significantly impacts their health behaviors and lifestyle; increasing health awareness can not only slow the progression of lifestyle-related diseases but also motivate continued improvement in health status [7]. Although existing studies have confirmed the correlation between health awareness and healthy lifestyles, few have delved into the underlying mechanisms of this relationship. Health awareness may be an important predictor of healthy lifestyles among university students. Therefore, Hypothesis 1 (H1) is proposed: Health awareness will significantly and positively influence university students' healthy lifestyles.

Social Cognitive Theory originates from learning theory and posits an interactional model between cognition, behavior, and environment [8]. This theory suggests that an individual's cognition and behavior interact with their environment and emphasizes the importance of cognitive factors in this interaction process [9]. Specifically, social cognitive theory asserts that human behavior is influenced not only by the external environment but also by internal cognitive processes. An individual's health awareness, as an internal cognitive drive, encourages them to adopt a healthy lifestyle [7]. This focus on health behaviors can enhance an individual's self-esteem, making individuals with strong self-esteem more likely to believe they can engage in healthy behaviors. Whether in adolescence or adulthood, self-esteem plays a crucial role in an individual's overall health [10]. There is a positive correlation between self-esteem and healthy lifestyles; individuals with high levels of self-esteem are more likely to engage in healthy behaviors, such as maintaining good dietary habits and regular exercise [11]. Thus,

Hypothesis 2 (H2) is proposed: Health awareness will have a significant indirect effect on healthy lifestyles through the mediation of self-esteem.

Social Support refers to mutual assistance behaviors among people, characterized by care, understanding, and recognition. It includes not only emotional support but also practical help. Social support can be obtained through various channels, such as family members, friends, colleagues, and community organizations [12]. Studies have shown a positive correlation between social support and healthy behaviors among adolescents. This means that receiving emotional and practical support from family, friends, or the community helps adolescents actively participate in physical activities and maintain a healthy lifestyle. Such support not only enhances their willingness to engage in exercise but also provides necessary resources and motivation, making it easier for them to stick to regular physical activities [13]. Parental and peer social support significantly influences adolescents' lifestyle behaviors. Specifically, the more support adolescents receive from parents and peers, the more actively they engage in physical activities and increase their intake of fruits and vegetables [14, 15]. An increase in health awareness can enhance individuals' interactions within their social support networks, thus promoting the implementation of healthy behaviors [16, 17]. Therefore, Hypothesis 3 (H3) is proposed: Health awareness will have a significant indirect effect on healthy lifestyles through the mediation of social support.

Health awareness encompasses a wide range of health-related knowledge, attitudes, and behaviors [18], including physical health, mental health, nutrition, exercise, and disease prevention. Research has shown that reproductive health awareness can positively predict an individual's level of self-esteem; adolescents with higher reproductive health awareness tend to exhibit higher levels of self-esteem [19]. At the same time, self-esteem is also positively correlated with healthy lifestyles; individuals with high levels of self-esteem are more likely to adopt a healthy lifestyle, such as maintaining a balanced diet, thereby improving their overall health [20]. When individuals have lower levels of self-esteem, they are more likely to engage in unhealthy behaviors such as disordered eating [21]. Furthermore, while there is no significant correlation between self-esteem and parental relationships, there is a strong link between self-esteem and perceived social support. In other words, while relationships with parents may have a limited impact on self-esteem, perceived social support from friends, colleagues, or the broader community plays an essential role in enhancing self-esteem [22]. Thus, Hypothesis 4 (H4) is proposed: Health awareness will significantly affect healthy lifestyles through the chain mediating effects of self-esteem and social support.

# Material and methods

## Participants

This study used a random sampling method to recruit university students from a university in Guangdong Province, China. First, we obtained the list of students from the university and compiled it into a database. Then, a random number table generated by a computer was used to randomly select potential participants from the list, ensuring that each student had an equal opportunity to be selected. The recruitment period lasted from March 23, 2024, to April 12, 2024. The research team sent invitations via email to randomly selected students, including a link to the online survey and detailed study information. All invitation emails clearly stated that participation was entirely voluntary and provided the option to withdraw from the study.

Participants were required to read and agree to an online informed consent form before beginning the questionnaire. This form detailed the purpose, procedures, data confidentiality measures, and potential impacts of the study. Only after consent was obtained could

participants access and complete the online questionnaire. The online survey platform ensured anonymity and data security to encourage honest responses.

A total of 1,297 students responded and completed the online questionnaire. To ensure data quality and validity, we performed a strict check and screening of the collected data based on the following criteria: (1) exclusion of questionnaires with invalid personal information, such as incomplete or contradictory answers; (2) exclusion of questionnaires with invalid responses, such as overly consistent answers or all questions answered with the same option. Ultimately, 1,169 participants (554 males and 615 females) were included in the study. The average age of participants was 20.56 years (SD = 1.73), with a range of 18 to 23 years.

Despite the convenience of online surveys in recruiting a large number of participants, there are some limitations. For example, online surveys may lead to sampling bias, as only students interested in online questionnaires or with time to participate may respond. Additionally, data collection online may reduce the quality and accuracy of questionnaire responses due to the lack of face-to-face interaction. To mitigate these limitations, the study took the following measures: Firstly, we minimized participants' concerns by clearly stating the voluntary nature of participation and data confidentiality, thereby improving response rates and authenticity. Secondly, we applied strict screening criteria during data analysis to eliminate invalid data and ensure the reliability of the final results.

## Measures

To ensure the applicability of the scales in a Chinese context, all scales used in this study underwent back-translation. Initially, the original English scales were translated into Chinese, and then another translator, unaware of the original content, translated them back into English. After comparison and adjustments, the final Chinese versions of the scales were confirmed.

**Health awareness.**   The Health Awareness Scale (HAS) [23] was used to measure the health awareness of university students. The scale consists of 6 items, such as "I reflect about my health a lot." It is scored on a 7-point Likert scale, ranging from 1 (strongly disagree) to 7 (strongly agree). Higher total scores indicate higher levels of individual health awareness. In this study, the Cronbach's alpha coefficient for this scale was 0.991.

**Healthy lifestyles.**   The Diet and Healthy Lifestyle Scale (DEVS) [24] was used to measure healthy lifestyles among university students. The scale consists of 14 items, such as "How many servings of whole grains do you consume in a day?" Each item is limited to three response options, ranging from 1 (minimal perceived intake) to 3 (maximum perceived intake). Higher total scores indicate a higher level of a healthy lifestyle. In this study, the Cronbach's alpha coefficient for this scale was 0.932.

**Self-esteem.**   The Rosenberg Self-Esteem Scale (RSES) [25] was used to measure self-esteem. The scale consists of 10 items, such as "I wish I could have more respect for myself." It is scored on a 4-point Likert scale, ranging from 1 (strongly disagree) to 4 (strongly agree). Higher total scores indicate higher levels of self-esteem. In this study, the Cronbach's alpha for this scale was 0.996.

**Social support.**   The Oslo Social Support Scale (OSSS-3) [26] was used to measure social support. The scale consists of 3 items, such as "How many people are so close to you that you can count on them if you have great personal problems?" The first item is rated on a 4-point Likert scale, while the second and third items are rated on a 5-point Likert scale. Higher total scores indicate higher levels of social support. In this study, the Cronbach's alpha for this scale was 0.981.

### Statistical analyses

Data were analyzed using the PROCESS macro in SPSS 27.0. The Harman single-factor test was conducted to prevent common method bias. Descriptive statistics and correlation analyses were performed to examine the relationships among health awareness, healthy lifestyles, self-esteem, and social support. To test the significance of the chain mediation model, model 6 in the PROCESS macro was used [27]. The bootstrap method was applied to test the significance of the mediation effect, with a confidence interval set at 95% and 5,000 iterations to eliminate the effects of multicollinearity.

### Ethical considerations

The study was approved by the Ethics Committee of Guangdong Business and Technology University on February 5, 2024 (approval number: 2024GS006). To ensure participants' privacy, the study was conducted anonymously. Before the study began, detailed informed consent was provided to all participants, and their written consent was obtained.

## Results

### Common method deviation test

The results of the Harman single-factor test showed that there were four factors with eigenvalues greater than 1, and the variance explained by the largest factor was 38.106% (less than 40%). Therefore, no significant common method bias was detected [28].

### Preliminary analyses

A total of 1,169 participants were included in this study, with 554 males (47.4%) and 615 females (52.6%). The average age of participants was 20.56 years (SD = 1.73), with an age range from 18 to 23 years. The mean of health awareness (HA) was 20.20 (SD = 7.713), healthy lifestyles (HL) was 22.847 (SD = 7.007), self-esteem (SE) was 24.502 (SD = 11.001), and social support (SS) was 6.824 (SD = 3.813). Descriptive statistics and correlation analysis results are shown in Table 1. Significant positive correlations were found among health awareness, healthy lifestyles, self-esteem, and social support.

### Chain mediation analyses

Chain mediation analysis results are shown in Fig 1, and regression analysis results are presented in Table 2. Health awareness significantly positively predicted healthy lifestyles ($\beta$ = 0.189, $p < 0.001$), self-esteem ($\beta$ = 0.145, $p < 0.001$), and social support ($\beta$ = 0.055, $p < 0.001$).

**Table 1. Means, standard deviations and correlation matrices of variables (*N* = 1169).**

|        | M      | SD     | 1       | 2       | 3       | 4 |
|--------|--------|--------|---------|---------|---------|---|
| 1.HA   | 20.200 | 7.713  | 1       |         |         |   |
| 2.HL   | 22.847 | 7.007  | .208*** | 1       |         |   |
| 3.SE   | 24.502 | 11.001 | .101**  | .384*** | 1       |   |
| 4.SS   | 6.824  | 3.813  | .122*** | .405*** | .124*** | 1 |

Notes

*$p < 0.05$

**$p < 0.01$

***$p < 0.001$. HA = Health Awareness; HL = Healthy Lifestyles; SE = Self-esteem; SS = Social Support. The following were the same.

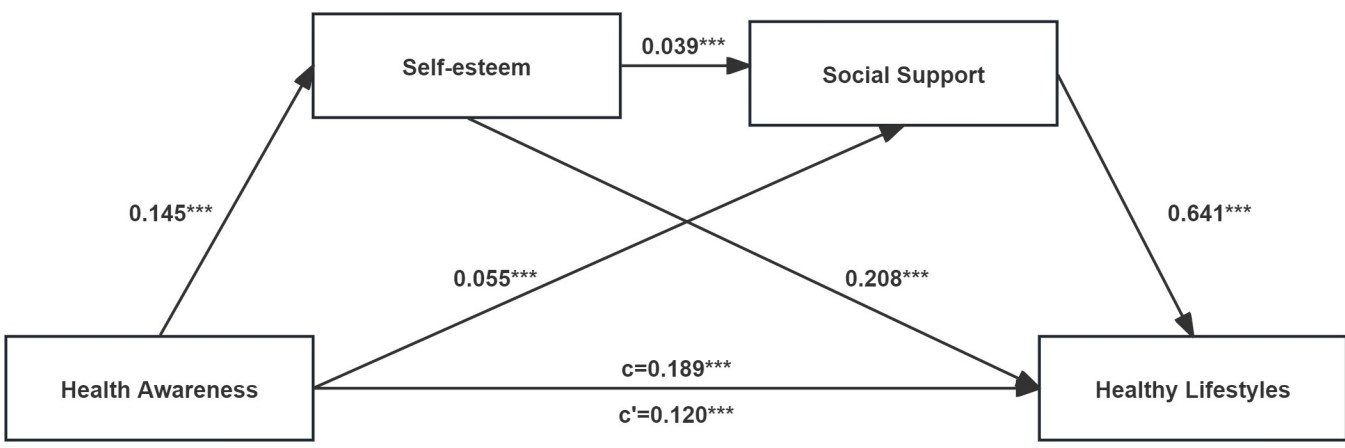

**Fig 1. The result of chain mediation model.** ***p < 0.001. The data of all variables were standardized.

Additionally, self-esteem significantly positively predicted healthy lifestyles ($\beta$ = 0.208, p < 0.001) and social support ($\beta$ = 0.039, p < 0.001), while social support significantly positively predicted healthy lifestyles ($\beta$ = 0.641, p < 0.001). After adding self-esteem and social support as mediating variables, the predictive effect of health awareness on healthy lifestyles remained significant ($\beta$ = 0.120, p < 0.001). These findings suggest that self-esteem and social support play a partial mediating role in the relationship between health awareness and healthy lifestyles.

The results of the mediation effect analysis are presented in Table 3. The mediation effect consists of three paths of indirect effects: Path 1: HA→SE→HL ($\beta$ = 0.030, 95% CI [0.013, 0.049]); Path 2: HA→SS→HL ($\beta$ = 0.035, 95% CI [0.017, 0.055]); Path 3: HA→SE→SS→HL ($\beta$ = 0.004, 95% CI [0.001, 0.007]). All three paths were found to be significant.

## Discussion

In this study, we constructed a chain mediation model among university students to explore the relationship between health awareness and healthy lifestyles, and the mediating roles of

**Table 2. Regression analysis between variables (N = 1169).**

| Dependent variables | Predictive variables | R | $R^2$ | F | $\beta$ | t | Boot 95% CI |
|---|---|---|---|---|---|---|---|
| **HL** | | 0.54 | 0.29 | 161.97*** | | | |
| | HA | | | | 0.12 | 5.32*** | [0.08,0.17] |
| | SE | | | | 0.21 | 13.14*** | [0.18,0.24] |
| | SS | | | | 0.64 | 13.97*** | [0.55,0.73] |
| **SS** | | 0.17 | 0.03 | 16.44*** | | | |
| | HA | | | | 0.06 | 3.81*** | [0.03,0.08] |
| | SE | | | | 0.04 | 3.88*** | [0.02,0.06] |
| **SE** | | 0.10 | 0.01 | 12.12*** | | | |
| | HA | | | | 0.15 | 3.48*** | [0.06,0.23] |

Notes

*p < 0.05

**p < 0.01

***p < 0.001. The data of all variables were standardized, the following were the same.

**Table 3. Indirect effect of HA on HL via SE and SS (N = 1169).**

| Effect | Product of Coefficients | | Boot95% CI | |
|---|---|---|---|---|
| | Effect | Boot Se | LLCI | ULCI |
| HA→ SE →HL | 0.030 | 0.009 | 0.013 | 0.049 |
| HA→SS→HL | 0.035 | 0.010 | 0.017 | 0.055 |
| HA→ SE →SS→HL | 0.004 | 0.002 | 0.001 | 0.007 |
| Direct effects: HA→HL | 0.120 | 0.023 | 0.076 | 0.165 |
| Total indirect effect | 0.069 | 0.014 | 0.041 | 0.097 |
| Total effect | 0.189 | 0.026 | 0.138 | 0.240 |

Notes: HA = Health Awareness; HL = Healthy Lifestyles; SE = Self-esteem; SS = Social Support.

self-esteem and social support in this context. The results indicate a significant positive correlation between health awareness and healthy lifestyles, and health awareness was found to significantly and positively predict healthy lifestyles (H1). Further analysis revealed that self-esteem serves as a mediator in the relationship between health awareness and healthy lifestyles (H2); similarly, social support also mediates this relationship (H3). This study further reveals that health awareness affects healthy lifestyles through the chain mediation effect of self-esteem and social support (H4).

## Findings

Firstly, the results of this study show a significant positive correlation between health awareness and healthy lifestyles among the participants. The stronger the health awareness of university students, the more likely they are to engage in positive health behaviors. A previous study on university students' healthy lifestyles found that students with a higher degree of concern for health are more likely to adopt healthy lifestyles. Compared to non-medical students, medical students have a higher cognitive understanding and perception of health information, which promotes their adoption of healthy lifestyles [29]. Other research has shown that consumers aware of the health benefits of consuming organic products are more willing to pay higher prices for these products. This suggests that health awareness can not only encourage healthier dietary choices but also influence consumer behavior [30]. The research framework of this study further expands on this theory, indicating that health awareness can also significantly predict other aspects of healthy lifestyles. According to the results of this study, enhancing health awareness among university students can directly promote their adoption of healthier lifestyles, thereby improving overall health levels.

Secondly, the study found that self-esteem mediates the relationship between health awareness and healthy lifestyles. Previous research has indicated that individuals with high levels of oral health behavior and self-esteem are closely associated with oral health awareness [31]. These studies emphasize the importance of health awareness in enhancing self-esteem, further supporting the findings of this study. Self-esteem is an individual's evaluation and perception of themselves; it is an essential component of self-cognition [32]. High self-esteem means an individual has a highly positive overall evaluation of themselves, while low self-esteem implies a more negative self-evaluation. Compared to individuals with low self-esteem, those with high self-esteem have better overall health status [33]. Self-esteem is one of the strongest predictors of eating disorders. People with eating disorders often have lower self-esteem. Individuals with low self-esteem are more likely to engage in binge eating and unhealthy dieting behaviors, which further demonstrates the crucial role of self-esteem in healthy lifestyles [34]. Self-esteem is positively correlated with positive health behaviors. Previous studies have also shown the

importance of self-esteem in understanding and promoting health behaviors [35]. These studies consistently indicate that people with high self-esteem are more likely to adopt and maintain healthy lifestyles.

Additionally, social support also plays a crucial mediating role in the impact of health awareness on healthy lifestyles. Parental and peer social support significantly influences adolescents' lifestyle behaviors; the more comprehensive this support, the more likely individuals are to engage in exercise and consume fruits and vegetables [36]. Moreover, such support, including encouragement, role modeling, and sharing health goals, has been shown to significantly improve individuals' healthy eating and exercise habits [37]. Enhancing an individual's cognitive beliefs about their ability to make healthy choices and engage in healthy behaviors can reduce the difficulty of execution and ultimately promote healthy behaviors [38]. This implies that enhancing health awareness can increase confidence in adopting healthy lifestyles, making it easier to engage in healthy behaviors. Furthermore, individuals with high self-esteem generally perceive themselves to have more social support, and this perception of social support is closely related to various health outcomes, including longevity [39]. Therefore, by improving university students' health awareness, their self-esteem and perceived social support can be enhanced, thereby improving their healthy lifestyles.

## Implications

This study proposed a chain mediation model to explore the process and mechanism by which the independent variable (health awareness) influences the dependent variable (healthy lifestyles) by examining the roles of two mediating variables (self-esteem and social support). The results show that health awareness indirectly promotes healthy lifestyles among university students by enhancing self-esteem and increasing social support. The final sample of this study included 1,169 participants, a large sample size that effectively reduces errors caused by small sample sizes and improves the reliability of the research results.

## Limitations

However, this study has several limitations. First, the study did not collect certain variables from respondents, such as family background, self-efficacy, and other factors. Research shows that self-efficacy is an important predictor of promoting healthy lifestyles [40]. Examining the relationship between health awareness and healthy lifestyles while controlling for these personal factors would make the research more in-depth and comprehensive. Second, since the respondents of this study were university students, the results may not be generalizable to other populations. However, studying the healthy lifestyles of young people is highly important, as the behavioral habits formed during their university years and adolescence may have a profound impact on their lifelong health practices [41]. Lastly, this study employed a cross-sectional design; therefore, future research should use longitudinal studies to explore changes in health awareness, self-esteem, social support, and healthy lifestyles among university students, to better understand the causal relationships between variables.

Despite these limitations, this study still has significant theoretical and practical implications. From a theoretical perspective, the study constructed a chain mediation model that reveals how health awareness influences university students' healthy lifestyles through self-esteem and social support, enriching the theoretical foundation of the relevant field and providing a reference framework for future research. From a practical perspective, the results offer valuable insights for enhancing healthy lifestyles among university students, serving as an important reference for educational institutions and policymakers in designing health education and support policies. Specifically, universities could implement mandatory physical

education courses to help students enhance their self-awareness and healthy behaviors [42]. Meanwhile, schools, communities, and other relevant departments should strengthen their support for students' health by providing more health resources and social support [43]. Through these measures, healthy lifestyles among university students can be effectively improved, promoting overall health levels.

## Conclusion

In conclusion, this study reveals the potential pathways through which health awareness affects healthy lifestyles among university students. The results show a significant positive relationship between health awareness and healthy lifestyles, and that health awareness can significantly predict the development of healthy lifestyles. Additionally, health awareness indirectly influences healthy lifestyles through the chain mediation effect of self-esteem and social support. These findings provide important empirical support for designing interventions to improve healthy lifestyles among university students.

## Acknowledgments

We would like to express our gratitude to all the teachers who provided help and guidance throughout this research. We also extend our thanks to all the students who participated in the survey.

## Author Contributions

**Conceptualization:** Zeqing Zhang, Akmar Hayati Ahmad Ghazali.

**Data curation:** Zeqing Zhang, Zerui Huang.

**Formal analysis:** Zeqing Zhang, Akmar Hayati Ahmad Ghazali.

**Investigation:** Jeffrey Lawrence D'Silva, Zerui Huang.

**Methodology:** Zeqing Zhang.

**Software:** Haslinda Abdullah, Ismi Arif Ismail.

**Supervision:** Haslinda Abdullah, Ismi Arif Ismail.

**Writing – original draft:** Zeqing Zhang.

**Writing – review & editing:** Zeqing Zhang.

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
