## [Decision Letter · Decision Letter 0]

5 Aug 2024

PONE-D-24-27865The Influence of Health Awareness on University Students' Healthy Lifestyles: The Chain Mediating Role of Self-esteem and Social SupportPLOS ONE

Dear Dr. Zhang,

Thank you for submitting your manuscript to PLOS ONE. After careful consideration, we feel that it has merit but does not fully meet PLOS ONE’s publication criteria as it currently stands. Therefore, we invite you to submit a revised version of the manuscript that addresses the points raised during the review process.

We look forward to receiving your revised manuscript.

Kind regards,

Nik Ahmad Sufian Burhan

Academic Editor

PLOS ONE

Journal Requirements:

Reviewers' comments:

Reviewer's Responses to Questions

**Comments to the Author**

1. Is the manuscript technically sound, and do the data support the conclusions?

Reviewer #1: Yes

Reviewer #2: Yes

2. Has the statistical analysis been performed appropriately and rigorously? 

Reviewer #1: Yes

Reviewer #2: Yes

3. Have the authors made all data underlying the findings in their manuscript fully available?

Reviewer #1: Yes

Reviewer #2: Yes

4. Is the manuscript presented in an intelligible fashion and written in standard English?

Reviewer #1: Yes

Reviewer #2: Yes

5. Review Comments to the Author

Reviewer #1: please check attachment

Areas for Improvement:

Clarify Research Objective: The research objective should be explicitly stated at the beginning of the paper.

Problem Statement: The problem statement needs more clarity. How was the hypothesis developed? Was it based on theoretical foundations or identified through a literature gap?

Mediator Role: Provide a detailed explanation of the mediator's role and its function within the context of this research.

Sample Selection Process: Elaborate on how the sample was selected.

Online Survey Process and Limitations:

Detail the process by which participants voluntarily participated and submitted their questionnaires through an online survey.

Discuss any limitations associated with using an online survey and how these were mitigated.

Instrument Translation: If the instrument was translated, describe the translation process and how the translated instrument reached the sample.

Recommendations:

Literature Review Section: Create a separate literature review section instead of including it in the introduction.

Discussion, Limitations, and Future Research: These sections should be thoroughly explained and clearly outlined.

Reviewer #2: for literature review, formulate the hypothesis into related literature review section. In the methodology section, add a paragraph if the instrument has been translated into Chinese version, add an example of item as well as add a demographic profiling of respondents. Refers to attachment file

6. PLOS authors have the option to publish the peer review history of their article (what does this mean?). If published, this will include your full peer review and any attached files.

Reviewer #1: **Yes: **Balan Rathakrishnan

Reviewer #2: No

---

## [Author Response · Author response to Decision Letter 0]

17 Sep 2024

Dear Academic Editor and Reviewers,

Thank you for reviewing our manuscript, "The Influence of Health Awareness on University Students' Healthy Lifestyles: The Chain Mediating Role of Self-esteem and Social Support." We have carefully considered all the feedback provided and have made the necessary revisions to address each point. Below is a detailed response to the suggestions provided:

Reviewer 1:

Clarify Research Objective: We have explicitly stated the research objective at the beginning of the paper to provide readers with a clearer understanding of the study's purpose and direction.

Problem Statement: We have clarified the problem statement in the introduction, explaining that the hypothesis was developed based on theoretical foundations and a gap identified in the literature.

Mediator Role: We have provided a detailed explanation of the mediator's role and its function within the context of this research in the literature review section.

Sample Selection Process: We have elaborated on the sample selection process in the "Participants" section, including details on how the random sampling method was implemented.

Online Survey Process and Limitations: We have detailed the process by which participants voluntarily participated and submitted their questionnaires through an online survey, and discussed any limitations associated with using an online survey, along with the measures taken to mitigate them.

Instrument Translation: We have added a description of the translation process of the instruments and how the translated instruments were applied to the sample.

Literature Review Section: As recommended, we have created a separate literature review section instead of including it in the introduction.

Discussion, Limitations, and Future Research: We have thoroughly explained and clearly outlined these sections in the revised manuscript.

Reviewer 2:

Abstract: We have added 1-2 sentences briefing on the study before stating the aims. Additionally, we have been more specific about the theoretical and practical implications.

Introduction: We have relocated each hypothesis to the related literature review section.

Participants Section: In the "Materials and Methods" section, we have elaborated on how the random sampling method was implemented.

Measures Section: We have described the back-to-back translation process involved in adopting the questionnaires and included 1-2 examples of items for each measure.

Results Section: We have added a descriptive analysis of the demographic profiling of participants in the Results section.

Common Method Deviation Test Section: We have included the necessary citations for the Common Method Deviation Test.

Journal Requirements:

Compliance with PLOS ONE Style Requirements: We have ensured that the manuscript meets PLOS ONE's style requirements, including those for file naming.

Data Availability Statement: We confirm that our manuscript contains all the raw data required to replicate the results of our study.

Reference List Review: We have carefully reviewed the reference list to ensure that it is complete and correct. We confirm that there are no citations of retracted papers, and therefore, no deletions or replacements were necessary. All references cited in the list are current and valid.

We hope these revisions have adequately addressed all your suggestions and have improved the overall quality of the manuscript. Thank you for your time and effort in reviewing our work.

Sincerely,

Zeqing Zhang

---

## [Editor Report · Decision Letter 1]

27 Sep 2024

The influence of health awareness on university students' healthy lifestyles: The chain mediating role of self-esteem and social support

PONE-D-24-27865R1

Dear Dr. Zhang,

We’re pleased to inform you that your manuscript has been judged scientifically suitable for publication and will be formally accepted for publication once it meets all outstanding technical requirements.

Kind regards,

Nik Ahmad Sufian Burhan

Academic Editor

PLOS ONE
---

## [Editor Report · Acceptance letter]

1 Oct 2024

PONE-D-24-27865R1 

PLOS ONE

Dear Dr. Zhang, 

I'm pleased to inform you that your manuscript has been deemed suitable for publication in PLOS ONE. Congratulations! Your manuscript is now being handed over to our production team.

Kind regards, 

on behalf of

Dr. Nik Ahmad Sufian Burhan 

Academic Editor

PLOS ONE